# IL3D: A Large-Scale Indoor Layout Dataset for LLM-Driven 3D Scene Generation

## Abstract

In this study, we present IL3D, a large-scale dataset meticulously designed for large language model (LLM)-driven 3D scene generation, addressing the pressing demand for diverse, high-quality training data in indoor layout design. Comprising 27,816 indoor layouts across 18 prevalent room types and a library of 29,215 high-fidelity 3D object assets, IL3D is enriched with instance-level natural language annotations to support robust multimodal learning for vision-language tasks. We establish rigorous benchmarks to evaluate LLM-driven scene generation. Experimental results show that supervised fine-tuning (SFT) of LLMs on IL3D significantly improves generalization and surpasses the performance of SFT on other datasets. IL3D offers flexible multimodal data export capabilities, including point clouds, 3D bounding boxes, multiview images, depth maps, normal maps, and semantic masks, enabling seamless adaptation to various visual tasks. As a versatile and robust resource, IL3D significantly advances research in 3D scene generation and embodied intelligence, by providing high-fidelity scene data to support environment perception tasks of embodied agents. The dataset and accompanying code will be publicly accessible on GitHub.

## 1 Introduction

3D indoor scene generation has emerged as a pivotal technology bridging embodied intelligence, smart home design, virtual reality interaction, and robotic environmental perception. Its core objective is to transform abstract spatial requirements into physically plausible and semantically coherent indoor layouts Yang et al. (2024c); Çelen et al. (2024); Zhou et al. (2025). In this domain, large language models (LLMs), leveraging their robust natural language understanding and reasoning capabilities, have become critical tools for driving scene generation. However, the precise modeling of indoor scenes by LLMs heavily relies on high-quality synthetic datasets Khanna et al. (2024); Kolve et al. (2017). These datasets must not only encompass diverse indoor scene types but also provide fine-grained information to support 3D perception tasks, including object geometry, semantic relationships, and multimodal representations. Such data is essential to address challenges like object overlap, boundary overflow, and other physical plausibility issues, as well as to meet the requirements of 3D point cloud analysis, semantic segmentation, and related perception tasks.

While existing mainstream indoor scene synthesis datasets Khanna et al. (2024); Fu et al. (2021) have advanced 3D perception research to some extent, they exhibit notable limitations. Some datasets focus primarily on accumulating furniture assets but lack sufficient coverage of diverse indoor scene types, making it challenging to support modeling for varied living spaces such as bedrooms, kitchens, or study rooms. Others provide interactive 3D scenes but lack fine-grained annotations, failing to deliver critical information such as object materials, poses, or spatial relationships, which leads to biases in LLMs' semantic understanding of indoor scenes. Additionally, certain datasets prioritize visual realism but fall short in supporting the data formats required for multimodal 3D perception tasks Chen et al. (2024), such as depth maps, normal maps, or semantic masks, thereby limiting their utility for training perception models and hindering the synergy between 3D perception and scene generation tasks.

Furthermore, 3D perception tasks for indoor scenes impose stringent requirements on the "functionality" and "diversity" of datasets. On one hand, datasets must ensure that indoor layouts adhere to real-world functional logic (e.g., appropriate placement of cabinets and appliances in kitchens or

beds and wardrobes in bedrooms) to enable perception models to understand the functional context of indoor scenes. On the other hand, datasets need to cover indoor scenes with varying areas and object densities to prevent biases that could degrade the generalization ability of perception models in complex scenarios, such as densely populated living rooms or compact bathrooms. These demands underscore the urgent need for a large-scale synthetic dataset tailored to indoor scenes that addresses the requirements of 3D perception tasks.

To address these challenges, this study introduces the IL3D dataset, a large-scale indoor layout dataset designed to meet the core needs of indoor scene generation and 3D perception. By integrating high-quality existing scene resources and supplementing them with targeted synthetic data, IL3D overcomes the shortcomings of current datasets in terms of indoor scene diversity, annotation completeness for 3D perception, and adaptability to multiple tasks. The dataset provides training samples closely aligned with real-world indoor environments for LLMs while offering multimodal data, including semantic point clouds, 3D bounding boxes, and multi-view RGB images, to support 3D perception tasks. This facilitates a synergistic improvement in both the quality of indoor scene generation and the accuracy of 3D perception, laying a robust data foundation for subsequent academic research and industrial applications related to indoor scenes. Our contributions can be summarized as follows:

- We propose the IL3D dataset, comprising 27,816 indoor layouts and 29,251 indoor object models, effectively addressing the data requirements of various 3D scene understanding tasks.
- The dataset includes instance-level natural language descriptions and supports multiple data formats (semantic point clouds, 3D bounding boxes, multi-view RGB images, depth maps, normal maps, and semantic masks), enabling compatibility with diverse downstream visual tasks.
- Experimental results demonstrate that direct supervised fine-tuning (SFT) on IL3D significantly enhances the performance of LLM-driven layout generation, underscoring its importance for generative tasks in indoor 3D scene synthesis.
- The dataset is primarily constructed using USDZ-format assets and USDA-format scenes, which can be directly parsed and analyzed by LLMs, ensuring compatibility with mainstream graphics processing and simulation software. We will publicly release this dataset along with comprehensive documentation.

## 2 RELATED WORK

The development of large-scale, high-fidelity indoor scene datasets is critical for advancing 3D scene understanding, embodied intelligence, and 3D scene generation, as these datasets provide diverse, semantically rich room and layout representations to support tasks such as object navigation, scene layout generation, and interactive simulation.

### 2.1 SYNTHESIS SCENE DATASET

3D-FRONT focuses on furniture resources, providing an asset library with 13,151 high-quality 3D object models and 6,813 synthetic houses Fu et al. (2021). Its modular object component design enables flexible scene assembly, laying a solid foundation for the subsequent construction of full-scene datasets.

Building on 3D-FRONT, 3D-FUTURE further enhances data adaptability by offering 9,938 high-quality 3D CAD furniture models with texture and semantic annotations. It supports the export of color images, semantic masks, depth maps, and normal maps, significantly expanding the dataset's application in visual tasks like scene synthesis and texture transfer.

AI2-THOR Kolve et al. (2017) integrates sub-datasets including iTHOR, RoboTHOR Deitke et al. (2020), ProcTHOR Deitke et al. (2022), and ArchitecTHOR Eftekhar et al. (2023), covering 120 large-scale interactive 3D indoor scenes (e.g., kitchens, bedrooms, bathrooms, and living rooms). With over 2,000 unique interactive objects rendered via the Unity engine, it supports visual question answering (VQA) Antol et al. (2015) and physics-based simulation experiments, providing critical support for embodied intelligence research.

HSSD offers 211 photorealistic 3D scenes based on real-world floor plans, containing 18,656 3D object assets Khanna et al. (2024). By balancing scene diversity and physical plausibility, it significantly improves the generalization ability of agents from simulated to real-world environments in object goal navigation tasks, outperforming many previous synthetic datasets.

CHOrD Su et al. (2025) achieves breakthroughs in scale and controllability, including 9,706 collision-free house-scale indoor scenes with hierarchical layouts and customizable floor plans. Through advanced optimization techniques for fine-tuning object placement and spatial organization, it enables fine-grained control over scene topology, serving as a robust foundation for digital twin generation and large-scale simulation research.

MetaScenes provides 10,245 high-fidelity indoor scenes with multimodal annotations (semantic, geometric, and natural language descriptions) and dynamic interaction support Yu et al. (2025). It optimizes scene realism and interactivity, making it particularly suitable for multimodal learning and virtual reality applications, and opens up new possibilities for 3D scene generation and embodied intelligence research.

## 2.2 TEXT IN 3D SCENE

Scan2Cap proposes an RGB-D scan-based context-aware dense annotation task for 3D point cloud scenes, aiming to predict bounding boxes and their natural language descriptions for objects Chen et al. (2021). Using an end-to-end training framework, it leverages an attention mechanism to generate descriptive text and a message-passing graph module to capture inter-object spatial relationship features. On the ScanRefer dataset **??**, it achieves a 27.61% improvement in CiDEr@0.5IoU score compared to 2D baseline methods, laying an important foundation for dense annotation of 3D scenes.

ScanQA Azuma et al. (2022) focuses on visual question answering (VQA) in 3D scenes, constructing open-ended question-answer pairs based on the ScanNet dataset Dai et al. (2017). These pairs cover tasks such as object recognition, spatial relationship understanding, and scene context comprehension. By fusing point cloud features with language prompts, ScanQA enables fine-grained understanding of complex 3D scenes, significantly improving the accuracy of VQA tasks.

SQA3D Ma et al. (2022) further expands the complexity of 3D scene QA by introducing spatial reasoning and multi-object interaction tasks. Based on RGB-D scans, it provides rich semantic and spatial relationship annotations, supporting diverse tasks ranging from single-object recognition to complex scene reasoning. Through explicit textual relationship prompts, it optimizes multi-object reasoning performance, offering new insights for language-guided 3D scene understanding.

ExCap3D Yeshwanth et al. (2025) proposes an expressive 3D annotation task to generate multi-level 3D object descriptions, including high-level object descriptions and low-level part attribute descriptions. Based on the ScanNet++ dataset Yeshwanth et al. (2023), it uses vision-language models to generate 190,000 multi-view annotations (covering 34,000 objects and 947 indoor scenes). By ensuring semantic consistency of constrained text and textual similarity in latent space, it significantly improves description quality, with CiDEr scores outperforming existing methods.

## 2.3 LLM-DRIVEN SCENE GENERATION

Research on text-driven scene generation converts complex scene layouts into structured textual descriptions and leverages the reasoning capabilities of LLM to achieve efficient scene generation.

LayoutGPT Feng et al. (2023) proposes a prompt-based generation framework that effectively processes complex language prompts containing numerical values and spatial relationships. This framework significantly improves the accuracy and semantic consistency of generated layouts. I-Design adopts a multi-agent system and scene graph generation mechanism, supporting users to generate and visually design targets through natural language interaction. It enables an iterative generation process, transforming user preferences into complete 3D layouts. Holodeck Yang et al. (2024c) focuses on language-guided 3D embodied AI environment generation. It can create diverse indoor scenes, capture the semantics of complex queries, and support zero-shot object navigation tasks. LayoutVLM Sun et al. (2025) integrates vision-language models with differentiable optimization, proposing an innovative scene layout representation method. It generates physically plausible and

Table 1: Statistics of synthetic datasets for indoor scenes.

| Dataset | Reference | Number of Rooms | Number of Assets | **Text Annotation** |
|---------|-----------|-----------------|------------------|---------------------|
| 3D-Front | CVPR 2021 | 21.3K | 16.6K | ✗ |
| HSSD | CVPR2024 | 1.1K | 18.6K | ✗ |
| IL3D (Ours) | - | 27.8K | 29.2K | ✓ |

semantically consistent layouts from unlabeled 3D assets and language instructions, ensuring the robustness of scenes in terms of physical stability and functionality.

LLplace Yang et al. (2024b) optimizes the spatial coherence and functional relationship prediction of LLMs through supervised fine-tuning. It performs exceptionally well in generating layouts for complex indoor environments, excelling in both practicality and semantic accuracy. OptiScene Yang et al. (2025) enhances the physical plausibility and visual consistency of generated results through two-stage fine-tuning (supervised fine-tuning followed by direct preference optimization). It performs particularly well in handling diverse room types and complex spatial constraints.

## 3 THE IL3D DATASET

### 3.1 DATA

The IL3D dataset integrates the 3D-FRONT and HSSD datasets, with manual cleaning performed to remove data with abnormal sizes or layouts. We adopted the method of HOLODECK to synthesize some types of scenes that are missing in these source datasets. The IL3D dataset can be divided into two parts overall: 3D assets and indoor scene layouts. Among them, 3D assets include corresponding object models of different types and multiple types of asset annotations, while scene layouts include information on the position, rotation, and scaling of each object in the scene, as well as the corresponding range of the room.

We constructed the dataset based on the Universal Scene Description (USD) format: using 3D object assets in usdz format and room layouts in usda format. The most prominent feature of this format is text readability, meaning large language models can directly read information about objects in the scene from 3D scene models; relevant details are provided in Appendix A.2.2. Considering that current synthetic scene datasets generally lack natural language annotations, in the IL3D dataset, we annotated each 3D object asset with multiple annotations at different levels. Specifically, coarse labels and fine labels refer to the category names of objects at different hierarchical levels. Additionally, we used Qwen3-VL to annotate scenes with more detailed instance-level descriptions based on multi-view images, including the specific type of the object, appearance description, approximate weight, original pose, constituent materials, and spatial relationships of the object's layout in the room.

### 3.2 METRICS

The evaluation metrics of the IL3D dataset are divided into two categories: objective metrics and subjective metrics, which are used to comprehensively measure the quality and practicality of generated scenes. Objective metrics include Out-of-Bound (OOB), which is used to detect whether objects in the scene exceed the room boundaries; Object Overlap Rate (OOR), which is used to evaluate the overlap degree between objects; Generation Success Rate (GSR), which is used to calculate the success rate of scene generation; and CLIP-Similarity (CLIP-Sim), which measures the semantic similarity between the generated scene and the reference scene based on the CLIP model.

Subjective metrics include GPT Ratings, where the GPT model is used to score the overall quality of generated scenes. These subjective scores cover the following aspects: Object Pose (OP), which evaluates the rationality of object poses; Physical Reality (PR), which checks the physical authenticity of the scene; Semantic Consistency (SC), which verifies the semantic consistency of objects in the scene; Scene Functionality (SF), which evaluates whether the scene meets practical functional requirements; and Visual Aesthetics (VA), which measures the visual aesthetics of the scene.

Objective metrics and subjective metrics together form a comprehensive and systematic framework for performance evaluation of the IL3D dataset, which is applicable to various tasks of 3D scene generation and scene understanding. The specific calculation methods are provided in Appendix A.4.3.

## 3.3 DATASET APPLICATIONS AND USAGE SCENARIOS

With its high-quality 3D assets, detailed scene layouts, and rich natural language annotations, the IL3D dataset provides important support for the fields of 3D scene understanding, generation, and interaction, meeting the diverse needs of academic research and industrial applications. The scene readability and multi-level annotations of the IL3D dataset make it an ideal resource for developing and evaluating 3D scene synthesis algorithms—especially for LLM-driven scene generation. After simple supervised fine-tuning, LLMs can generate complex indoor scenes with semantic consistency and physical authenticity using this dataset.

In addition, the high-fidelity scenes provided by the dataset are compatible with multiple simulation platforms, offering key support for robot navigation and interaction tasks. Researchers can use IL3D to train models, enabling robots to perform path planning, object grasping, or scene understanding in virtual environments, thereby simulating real-world indoor interaction scenarios. The natural language annotations of IL3D also provide unique opportunities for multimodal learning tasks, allowing researchers to develop vision-language models for indoor scenes and explore cross-modal correlations between 3D scenes and natural language.

## 4 DATASET ANALYSIS

### 4.1 ROOM TYPE

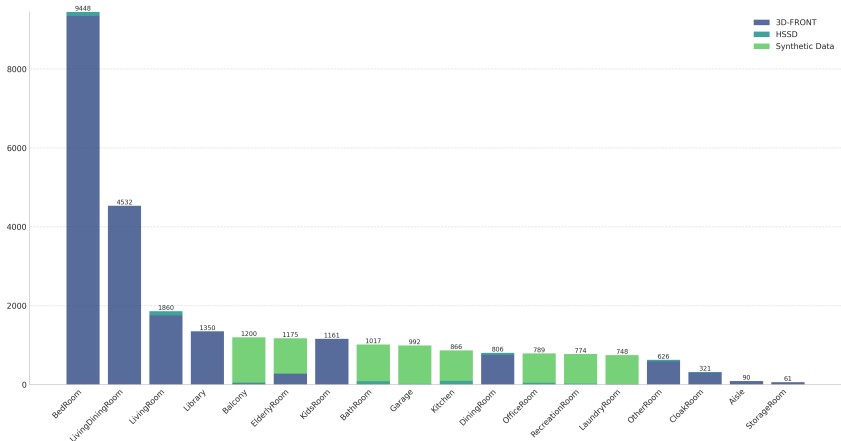

Figure 1: **Statistics of room type.** IL3D comprises 27,816 distinct rooms, consisting of 18 common room types. For existing datasets, we selectively supplemented some missing room types.

As shown in Fig. 1, we analyzed the room type distribution of the dataset, and as illustrated in the Fig. 1, this distribution reflects data integrated from the 3D-FRONT and HSSD datasets, supplemented by our targeted additional synthetic data. Among the room types, the bedroom category occupies a dominant position, with 9,148 samples, while living rooms and dining rooms follow closely, with 4,532 and 1,860 samples respectively. These scenes are primarily sourced from the 3D-FRONT dataset, underscoring its significant role in this field.

Notably, our synthetic data (highlighted in green) plays a crucial role in addressing underrepresented categories in other datasets. It enhances the diversity and balance of the dataset, thereby alleviating the lack of scenes belonging to certain specific categories in these existing datasets. The HSSD dataset, limited by its data scale, makes a relatively small contribution to scene layouts.

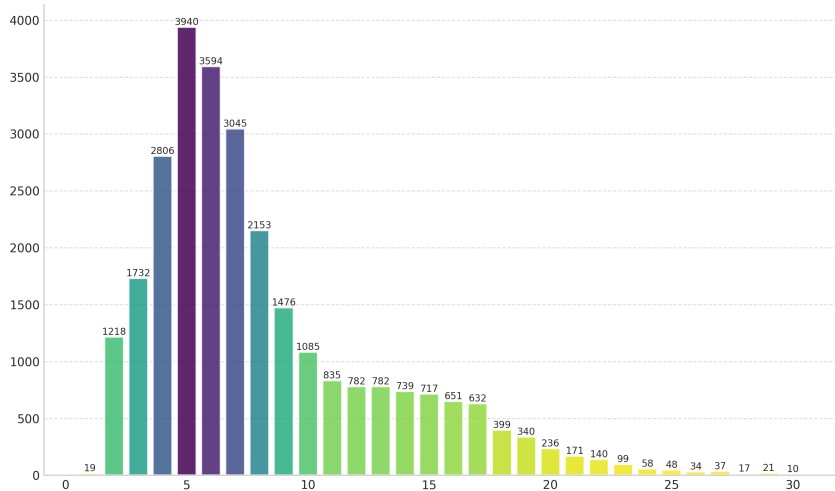

Figure 2: **Statistics of object numbers per room.** The number of objects in a room is mostly concentrated in the range of 4 to 9. When this range is exceeded, the corresponding number of rooms shows a steady downward trend as the number of objects increases.

Overall, the IL3D dataset contains over 27,000 samples, covering 18 room types. Among these samples, synthetic data accounts for approximately 20–30% of the total. This not only strategically expands the coverage of indoor scene categories but also fosters the robustness of LLM-driven scene generation and other visual tasks by reducing potential biases in model training.

## 4.2 NUMBER OF OBJECTS IN ROOMS

As shown in the Fig. 2, we counted the distribution of rooms with different numbers of objects. It can be observed that the number of objects in rooms within the dataset mainly ranges from 4 to 9. Rooms containing 5 objects are the most common, with a count of 3,594; this number primarily corresponds to the most typical indoor room types, such as living rooms and bedrooms.

Starting from rooms with 10 objects, the number of corresponding rooms drops below 1,000 and decreases steadily afterward, which reflects a good balance in the layout of moderately complex rooms. Even when the number of objects exceeds 20 (with the number of rooms falling below 200), there are still a considerable number of highly complex scenes in IL3D—this highlights the comprehensiveness of our dataset.

## 4.3 ROOM AREA

The area distribution of various room types in our indoor scene layout dataset reveals rich patterns of spatial features, as shown in Fig. 3. Fig. 3 (a) presents the density distribution of the total room area; most room types exhibit a multi-peak feature, reflecting the standardization of architectural design and the diversity of functional requirements. For example, the distribution of bathrooms and kitchens shows a significant peak around 5–15 square meters, embodying their compact design that prioritizes practicality; in contrast, larger spaces such as living rooms and garages have a wider distribution, with peaks exceeding 20 square meters, adapting to diverse furniture layouts and vehicle storage needs.

In comparison, Fig. 3 (b) shows the distribution of navigable area—a metric that accounts for obstacle-free regions used for movement. Notably, these distributions are shifted leftward compared to the total area, with reduced peak density and a narrowed range, highlighting the significant impact of fixed facilities, furniture, and built-in elements on non-navigable areas. For instance, the navigable area of kitchens and dining rooms is reduced by approximately 20–30% compared to their total area, which may be attributed to cabinets and appliances; meanwhile, open spaces such as atriums and entertainment rooms maintain a high proportion of navigable area.

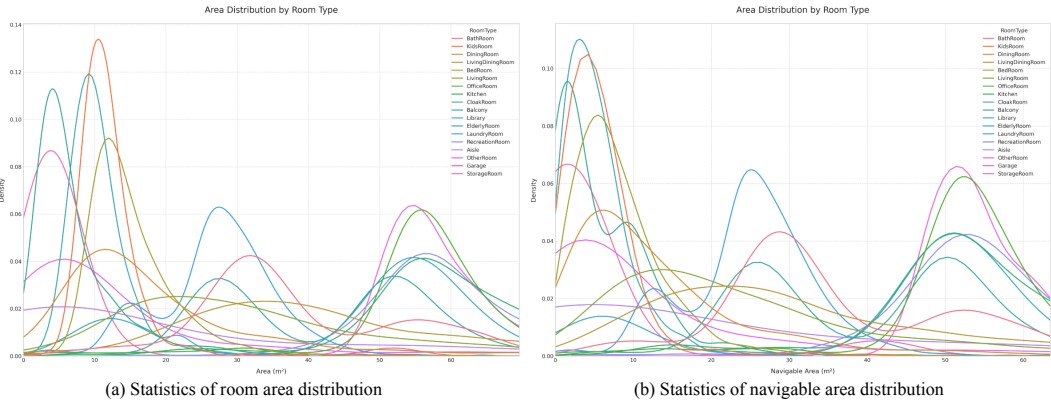

(a) Statistics of room area distribution   (b) Statistics of navigable area distribution

Figure 3: **Room Area Statistics.** Navigable areas generally exhibit a certain degree of deviation compared with room areas.

This difference underscores the value of the dataset in robotics and virtual reality applications: as a core metric for path planning and accessibility evaluation, navigable area provides critical references. In general, the total area reflects the functional attributes of a room, while the navigable area offers a more appropriate perspective for understanding human-centered spatial practicality in indoor environments.

## 4.4 OBJECT CATEGORIES

We counted object category distribution in the 3D asset library to understand the composition and diversity of common indoor elements. The pie chart in Fig. 4 shows the dataset's object categories hierarchically grouped into major types (Furniture, Lighting, Accessories, Others), each split into specific subcategories. For example, Furniture subcategories like Armchairs and Multi-Seat Sofas make up a big share—key for spatial functions; Lighting includes Pendant Lamps and Table Lamps, meeting different lighting needs. This hierarchy not only clarifies the modular nature of object categories but also reveals large proportional differences. For instance, Furniture accounts for over 70%, far more than others—due to the dataset focusing on residential and office indoor scenes. Overall, the distribution shows object categories are function-focused: high-frequency items (e.g., commonly used furniture) dominate, while low-frequency ones add realism and diversity to scenes.

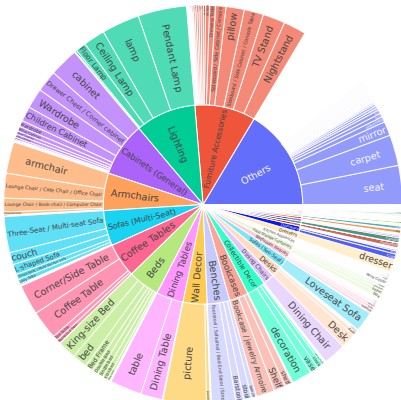

Figure 4: Statistics of object category distribution.

## 5 EXPERIMENTS

We investigated the impact of different dataset scales and scene data settings on LLM-Driven scene generation tasks, focusing on the performance of the same LLM after Supervised Fine-Tuning (SFT) under varying training data conditions.

## 5.1 EXPERIMENTAL SETUP

The scene generation process was divided into two stages: 3D asset retrieval and scene layout generation. To study the influence of natural language annotations on the spatial reasoning ability of LLMs, we adopted a "retrieval-then-generation" strategy—first retrieving object information in the scene based on text descriptions, then using the SFT-tuned LLM for reasoning and generation.

Table 2: **Main experimental results.** Comparison of Performance in objective and subjective metrics across I-Design, Holodeck and Qwen3-14B (Supervised Fine-Tuning on IL3D).

| Methods | Reference | Objective Metrics | | | | Subjective Metrics | | | | |
|---|---|---|---|---|---|---|---|---|---|---|
| | | OOB↓ | OOR↓ | CLIP-Sim↑ | GSR↑ | OP↑ | PR↑ | SC↑ | SF↑ | VA↑ |
| I-Design | ECCV 2024 | 39.883 | 17.466 | 26.406 | 37.8% | **7.417** | **8.767** | **7.467** | **6.5** | **7.317** |
| Holodeck | CVPR 2024 | **1.783** | **0.113** | 25.164 | 91.7% | 6.65 | 8.2 | 6.417 | 5.85 | 6.767 |
| Qwen3-14B | - | 16.455 | 4.494 | **27.357** | **100%** | 6.683 | 7.967 | 5.783 | 5.467 | 6.533 |

The test data consisted of 30 natural language descriptions for each of six randomly selected common room types. GPT-4o was used as the scoring model for subjective metrics. Qwen3 models with different parameter sizes were all fine-tuned via LoRA (Parameter-Efficient Fine-Tuning). Detailed experimental configurations are provided in Appendix A.2.

## 5.2 EXPERIMENTAL RESULTS

I-Design and Holodeck were selected as baselines to compare the performance of our model with other LLM-Driven scene generation methods.

## 5.3 MAIN RESULTS

As shown in Tab. 2, we compared the 3D scene generation performance of I-Design, Holodeck, and our SFT-tuned Qwen3-14B. In terms of objective metrics, Qwen3-14B exhibited excellent performance: its Out-of-Bound (OOB, 16.455) and Object Overlap Rate (OOR, 4.494) were much lower than those of I-Design, while its CLIP-Similarity (CLIP-Sim, 27.357) and Generation Success Rate (GSR, 100%) were higher than both baselines. This demonstrates the advantages of the SFT method in generation reliability and semantic alignment, which is attributed to the fact that one-time forward reasoning avoids the cumulative reasoning errors of agent-based methods.

In terms of subjective metrics, I-Design outperformed Qwen3-14B in Object Pose (OP), Physical Reality (PR), Semantic Consistency (SC), Scene Functionality (SF), and Visual Aesthetics (VA). Holodeck was close to I-Design in PR but slightly inferior in other metrics, while Qwen3-14B showed weaker performance in SC and SF due to the lack of iterative optimization.

In summary, the SFT method is efficient and stable, making it suitable for rapid scene generation; agent-based methods improve subjective quality through iterative optimization, but their stability needs further enhancement.

## 5.4 ABLATION STUDY

### 5.4.1 IMPACT OF DATASET SCALE

As shown in Tab. 3, we evaluated the 3D scene generation performance of the Qwen3-1.7B model after SFT on the HSSD, 3D-Front, and IL3D datasets, where the dataset scales increased in the order of HSSD < 3D-Front < IL3D.

In the unannotated reasoning scenario, IL3D achieved the best performance in OOB (5.684) and OOR (4.372), outperforming HSSD (OOB: 4.039, OOR: 36.978) and 3D-Front (OOB: 20.577, OOR: 5.040). It also ranked among the top in CLIP-Sim (26.498) and GSR (100%). These results indicate that IL3D, with its larger scale and higher diversity, exhibits excellent performance in geometric control and semantic alignment.

In the annotated reasoning scenario, HSSD's OOB metric decreased significantly (53.448), which may be due to its small scale making it difficult to adapt to complex annotation requirements. In contrast, IL3D (OOB: 16.505) and 3D-Front (OOB: 27.063) still maintained relatively good performance. The GSR of all datasets was close to 100%, and their CLIP-Sim values were slightly higher than those in the unannotated reasoning scenario.

Overall, IL3D demonstrated higher stability and superiority in both unannotated and annotated reasoning scenarios, an advantage mainly attributed to its large-scale and diverse dataset design.

Table 3: Comparison of Performance in Objective Indicators for Supervised Fine-Tuning with Qwen3-1.7B on Datasets of Different Scales.

| Models | OOB↓ | OOR↓ | CLIP-Sim↑ | GSR↑ |
|---|---|---|---|---|
| | Unannotated Inference | | | |
| HSSD | **4.039** | 36.978 | 24.918 | 98.9% |
| 3D-Front | 20.577 | 5.040 | 26.484 | 100% |
| IL3D | 5.684 | **4.372** | **26.498** | **100%** |
| | Annotated Inference | | | |
| HSSD | 53.448 | 35.875 | 24.998 | 100% |
| 3D-Front | 27.063 | 6.811 | **26.912** | 100% |
| IL3D | **16.505** | **5.551** | 26.774 | **100%** |

Table 4: Comparison of Performance in Objective Indicators for Supervised Fine-Tuning of Different Qwen3 Models on the IL3D Dataset.

| Datasets | OOB↓ | OOR↓ | CLIP-Sim↑ | GSR↑ |
|---|---|---|---|---|
| | Unannotated Inference | | | |
| Qwen3-4B | 34.560 | 6.513 | 26.498 | 100% |
| Qwen3-8B | 24.945 | **4.107** | 27.007 | 97.8% |
| Qwen3-14B | **17.479** | 5.391 | **27.013** | **100%** |
| | Annotated Inference | | | |
| Qwen3-4B | 27.996 | 5.809 | 27.193 | 100% |
| Qwen3-8B | 21.998 | 4.695 | 26.997 | 100% |
| Qwen3-14B | **16.455** | **4.494** | **27.357** | 100% |

### 5.4.2 IMPACT OF NATURAL LANGUAGE ANNOTATIONS

As shown in Tab. 4, we evaluated the performance of the Qwen3 model series (with parameter sizes of 1.7B, 4B, 8B, and 14B) on the IL3D dataset under both unannotated and annotated reasoning conditions.

In the unannotated reasoning scenario, Qwen3-1.7B exhibited good boundary control, with an OOB value of 5.684, an OOR value of 4.372, a CLIP-Sim value of 26.498, and a GSR of 100%, though its semantic similarity was relatively low. Qwen3-4B showed a tendency for boundary overflow, as its OOB value rose to 34.560, with an OOR of 6.513, a CLIP-Sim of 27.066, and a GSR that dropped to 94.4%, indicating that medium-sized models are prone to such boundary issues. Qwen3-8B and Qwen3-14B, as large-sized models, demonstrated better generation stability: their OOB values decreased to 24.945 and 17.479 respectively, their OOR values were 4.107 and 5.391, their CLIP-Sim values remained stable at 27.007 and 27.013, and their GSR values rebounded to 97.8% and 100%.

Annotated reasoning significantly improved the performance of all models, with the GSR of each model reaching 100%. For Qwen3-1.7B, the OOB value increased to 16.505, while the OOB values of the other three models (4B, 8B, 14B) decreased to 27.996, 21.998, and 16.455 respectively. The OOR value of Qwen3-1.7B rose to 5.551, whereas the OOR values of the other three models dropped to 5.809, 4.695, and 4.494. The CLIP-Sim values of all models fluctuated between 26.774 and 27.357, which indicates that natural language annotations mainly play a role in optimizing geometric consistency. These results further show that annotated reasoning effectively alleviates the boundary overflow problem in medium-sized and large-sized models.

In summary, the experiment revealed a trade-off between model scale and generation performance: small-sized models (e.g., Qwen3-1.7B) have high efficiency in boundary control but limited ability to capture semantic information; large-sized models (e.g., Qwen3-8B, Qwen3-14B) can optimize OOB and OOR metrics with the assistance of annotations, but their improvement in CLIP-Sim (semantic similarity) remains limited.

## 6 CONCLUSIONS

This study constructs the IL3D dataset and applies it to LLM-Driven indoor scene generation, establishing a new benchmark in the field of 3D scene generation. IL3D not only provides diverse indoor layouts and high-precision object assets but also adapts well to visual tasks such as 3D scene generation and editing, serving as a powerful tool for complex visual tasks and embodied intelligence research. Experiments verify its significant advantages in improving model performance and adaptability. Despite its limitation in scene-level relationship descriptions, IL3D's openness and extensibility lay a foundation for future exploration of deeper scene understanding.

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

# A APPENDIX

## A.1 LLM USAGE STATEMENT

In the process of experimental implementation and paper writing of this study, the usage of large language models (LLMs) is stated as follows:

In the experimental section, given that this work focuses on the LLM-Driven indoor scene generation task, we adopted Qwen3 series models (including versions with 1.7B, 4B, 8B, and 14B parameters) for supervised fine-tuning (SFT) to verify the effectiveness of the proposed IL3D dataset and related methods. We further combined the LoRA (Low-Rank Adaptation) parameter-efficient fine-tuning technique to optimize the model's spatial reasoning capability, which supported the implementation of core experimental links such as scene generation performance evaluation and ablation studies.

During the paper writing phase, we used LLM tools for manuscript polishing, mainly including optimizing the accuracy of academic expressions and organizing sentence logic to conform to the writing norms of scientific research papers. However, the core ideas, research design, experimental data and result analysis, conclusion derivation, and other key contents of the paper were independently completed by the authors. We strictly ensured the originality of all academic viewpoints and the authenticity of data, in full compliance with academic research ethics and normative requirements.

## A.2 EXPERIMENTAL DETAILS

### A.2.1 EXPERIMENT CONFIGURATION

Table 5: Training Configuration for Different Qwen3 Parameter Sizes

| Config | Qwen3-1.7B | Qwen3-4B | Qwen3-8B | Qwen3-14B |
|---|---|---|---|---|
| Optimizer | AdamW | AdamW | AdamW | AdamW |
| Learning Rate | 1e-4 | 5e-5 | 5e-5 | 5e-5 |
| LoRA Rank | 8 | 8 | 8 | 8 |
| LoRA Alpha | 32 | 32 | 16 | 16 |
| Target Models | all-linear | all-linear | all-linear | all-linear |
| Warmup Ratio | 0.05 | 0.05 | 0.05 | 0.05 |
| Batch Size | 2 | 2 | 2 | 2 |
| Max Length | 4096 | 4096 | 4096 | 4096 |
| GPU device | NVIDIA L40S | NVIDIA L40S | NVIDIA L40S | NVIDIA L40S |
| Training Time | $\sim$ 4h | $\sim$ 5h | $\sim$ 7h | $\sim$ 12h |

As shown in Tab. 5, we provide the detailed configuration information of the experiment, including parameter settings, hardware conditions, and training duration.

### A.2.2 3D ASSET RETRIEVAL

As shown in the Fig. 5, this study implements 3D asset retrieval based on the instance-level annotations of the IL3D dataset, primarily employing Qdrant to construct a vector database of text descriptions. This vector database maps each 3D asset to a vector representation corresponding to its instance-level annotations (e.g., descriptions of object type, appearance, and material), laying the data foundation for subsequent similarity-based retrieval.

During the text-driven scene generation process, a large language model (LLM) is first used to extract specific descriptions of the objects required in the scene from the text description of the target scene; these descriptions cover core information such as object category, key features, and functional attributes. The extracted object descriptions are then converted into vector representations, and the Qdrant vector database is queried based on this vector to find the text description with the highest semantic similarity—thereby retrieving the corresponding 3D asset. This ensures that the

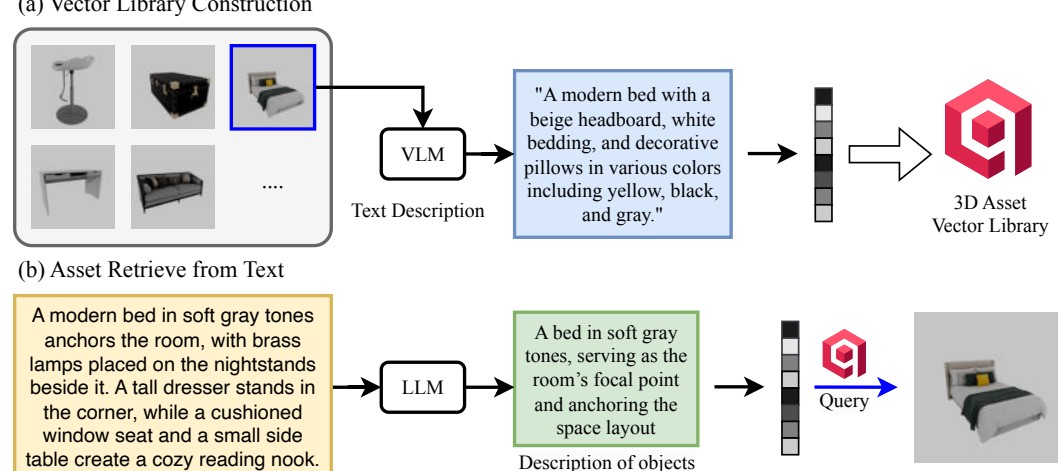

Figure 5: Asset Retrieval. (a) A 3D asset vector database for object descriptions is constructed based on Qdrant. (b) When generating scenes, LLMs (Large Language Models) are used to extract object information from text descriptions and retrieve relevant assets.

selected asset aligns with the semantic intent of the scene's text description, providing a reliable asset foundation for subsequent scene layout generation.

## A.3 SUPPLEMENTARY RELATED WORK

Image-guided scene generation focuses on extracting 3D spatial information from input images, typically implemented via end-to-end generative models or frameworks composed of multiple visual models. This direction enables scene reconstruction or synthesis directly from 2D image cues, providing critical support for tasks such as virtual reality, robotic perception, and 3D content creation.

CAST Yao et al. (2025) proposes a component-aligned 3D scene reconstruction method from a single RGB image. It first extracts object-level 2D segmentation and relative depth information, then leverages foundation models to infer 3D geometry and semantic relationships. This approach achieves high-fidelity scene recovery across diverse indoor and outdoor settings, laying a foundation for applications in virtual reality and robotics that require accurate 3D scene representations.

Gen3DSR Ardelean et al. (2024) is a generalizable 3D scene reconstruction framework that follows a "divide-and-conquer" strategy. It first processes the scene globally to extract depth and semantic information, then performs hierarchical optimization for object-level reconstruction. By generating compositional 3D scenes from single-view images, it demonstrates strong generalization capabilities to unseen environments, addressing the challenge of limited adaptability in traditional reconstruction methods.

DiffScene Xu et al. (2025) adopts a denoising diffusion model for generative indoor scene synthesis. Through a scene configuration denoising mechanism, it generates diverse and photorealistic 3D indoor environments from a single image or partial input. It supports key tasks including scene completion, object arrangement, and text-conditioned synthesis, and uses an unordered object set representation to enhance generation flexibility and applicability to downstream tasks.

MIDI Huang et al. (2025) extends a pre-trained image-to-3D object generation framework via a multi-instance diffusion model. It generates compositional 3D scenes from a single image, capable of handling the geometry and texture of multiple instances simultaneously in a single feed-forward pass. This efficiency makes it well-suited for rapid indoor layout synthesis, where multiple objects need to be integrated coherently.

SceneGen Meng et al. (2025) is a single-image 3D scene generation model that synthesizes geometry, texture, and relative poses in a single feed-forward pass—reportedly the first framework to

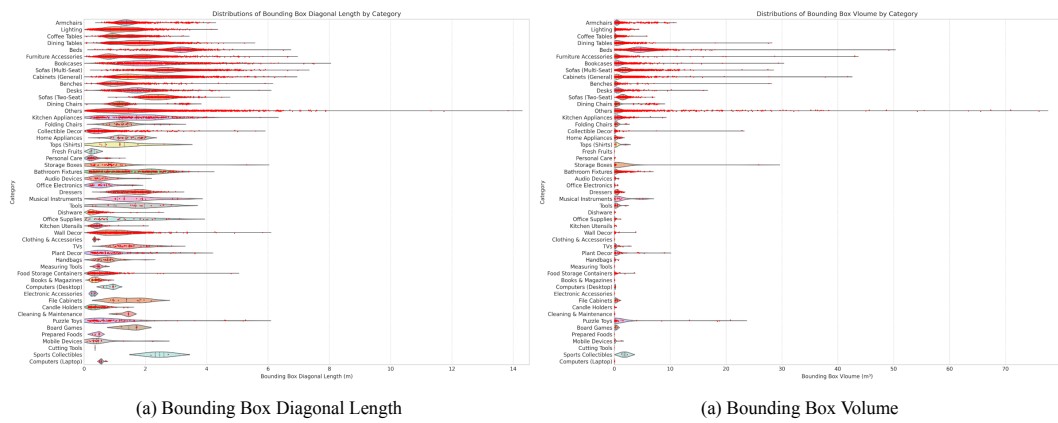

(a) Bounding Box Diagonal Length                    (a) Bounding Box Volume

Figure 6: Statistical Results of the physical scale distribution of assets.

achieve this capability. By streamlining the generation process, it enables fast construction of photorealistic indoor environments, reducing the computational overhead of multi-step scene building.

PhyScene Yang et al. (2024a) emphasizes the generation of physically interactive 3D scenes, designed specifically for embodied AI tasks. It creates indoor environments with realistic layouts and rich interactivity by incorporating physical constraints on object placement and considering articulated object interactions. This focus on physical plausibility ensures that the generated scenes can support practical embodied interactions (e.g., robot object manipulation), bridging the gap between synthetic scenes and real-world usability.

### A.4 SUPPLEMENTARY DATA STATISTICS

#### A.4.1 PHYSICAL SCALE OF THE ASSET LIBRARY

This section presents statistical results on the physical dimensions of 70 object categories in the 3D asset library, focusing on two key metrics: bounding box diagonal length and bounding box volume.

As show in Fig. 6 (a), for the bounding box diagonal length distribution (unit: meters), the distribution plot shows the diagonal length of instances in each category. Small-sized categories—such as fresh fruits, French fries, and food measuring tools—exhibit narrow distributions, indicating their consistent and compact dimensions, which are typical of desktop or handheld items. In contrast, large furniture categories (e.g., sofas, beds, and cabinets) display wide distribution ranges, reflecting size variations from compact to spacious designs. This characteristic is critical for 3D scene understanding algorithms, as it enables them to handle spatial layouts of multiple scales.

As show in Fig. 6 (b), for the bounding box volume distribution, all 70 categories show a similar variation trend, and their sorting order is consistent with that of the bounding box diagonal length—highlighting the correlation between volume distribution and linear dimensions in 3D assets. Categories with high morphological variability (e.g., office chairs, storage racks, and countertops) exhibit broad volume distributions, which reflects the impact of modular or customizable designs on occlusion and interaction modeling. In contrast, categories such as beverages, computers, and musical instruments have narrow volume distributions, emphasizing their standardized dimensions in the real world. This standardization facilitates accurate semantic segmentation and functional prediction in indoor 3D perception systems.

#### A.4.2 QUANTITY OF DIFFERENT ASSET CATEGORIES

The bar chart (shown in the Fig. 7) presents the quantity distribution of various object categories, sorted in descending order of quantity, highlighting the dominant status of certain objects in the dataset. For example, the "Others" category—comprising daily miscellaneous items such as seats and carpets—ranks first with over 4,000 instances, which reflects the universality and functional necessity of these objects in daily indoor spaces. It is followed by furniture categories such as sofas and tables, whose quantities gradually decrease from common to rare types until approaching zero.

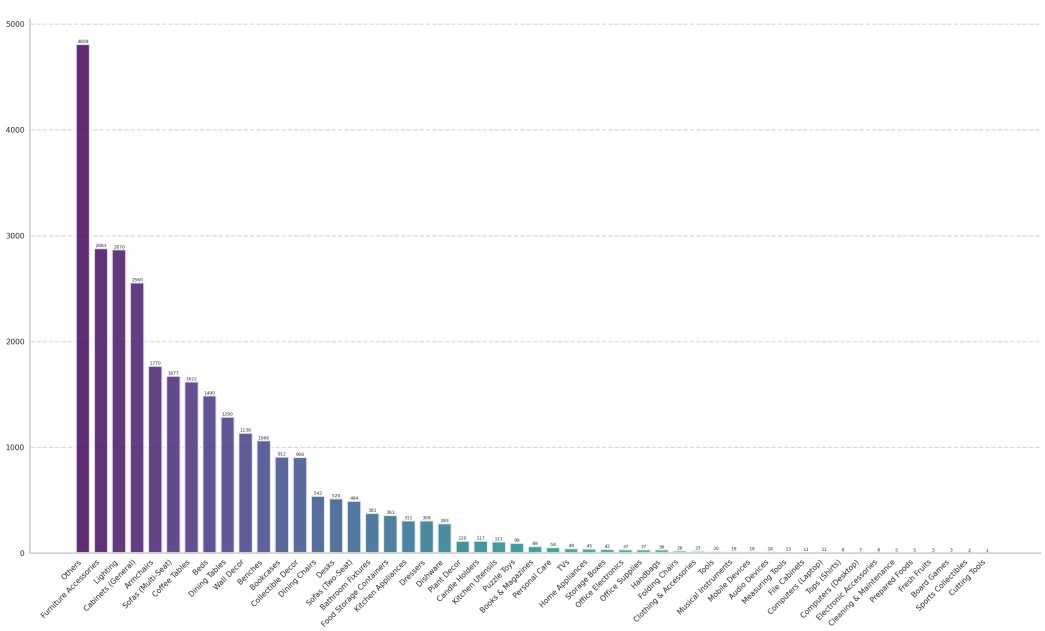

Figure 7: Statistics of different asset categories.

This long-tailed distribution indicates that the dataset captures the frequency pattern of objects in the real world: a small number of high-frequency objects (e.g., seating and storage items) dominate, while a large number of low-frequency objects (e.g., specific decorative items) contribute to diversity. This characteristic embodies the hierarchical and personalized features of indoor design.

### A.4.3 EVALUATION METRICS

**Out-of-Bound (OOB)** The Out-of-Bound (OOB) metric evaluates whether objects in a generated indoor scene exceed room boundaries (e.g., walls, floors, or ceilings). It ensures the physical plausibility of scenes, preventing irrational cases like objects floating or penetrating boundaries—an essential requirement for applications such as virtual reality, robotic simulation, and architectural design. A lower OOB value implies the generated scene better adheres to spatial constraints, thus enhancing realism and usability.

$$\text{OOB} = \frac{1}{N} \sum_{i=1}^{N} \mathbb{I}(V_{i,\text{out}} > 0) \tag{1}$$

where N is the total number of objects in the scene, $V_{i,\text{out}}$ denotes the volume of the i-th object that exceeds room boundaries, and $\mathbb{I}(\cdot)$ is an indicator function (taking 1 if the condition holds, 0 otherwise).

**Object Overlap Rate (OOR)** The Object Overlap Rate (OOR) quantifies the overlap degree between objects in a generated scene. In indoor scenarios, object overlaps (e.g., a chair penetrating a table) create unnatural layouts and reduce realism. OOR measures such collisions or overlaps to optimize generative models for physically reasonable object placement.

$$\text{OOR} = \begin{cases} \frac{\sum_{i=1}^{N} \sum_{j=i+1}^{N} V_{i,j}^{\text{inter}}}{\sum_{i=1}^{N} V_i} & \text{if } \sum_{i=1}^{N} V_i \geq 10^{-9}, \\ 0 & \text{otherwise} \end{cases} \tag{2}$$

where $V_{i,}$ is the bounding box volume of the i-th object, and $V_{i,j}^{\text{inter}}$ is the intersecting volume of the i-th and j-th objects' bounding boxes (only i¡j is computed to avoid duplicate counting of object pairs). If the total volume of all objects is extremely small, the OOR is set to 0 to avoid division errors by zero.

**Generation Success Rate (GSR)** The Generation Success Rate (GSR) measures the proportion of successfully generated scenes, i.e., whether the model can output valid scene results (as opposed to

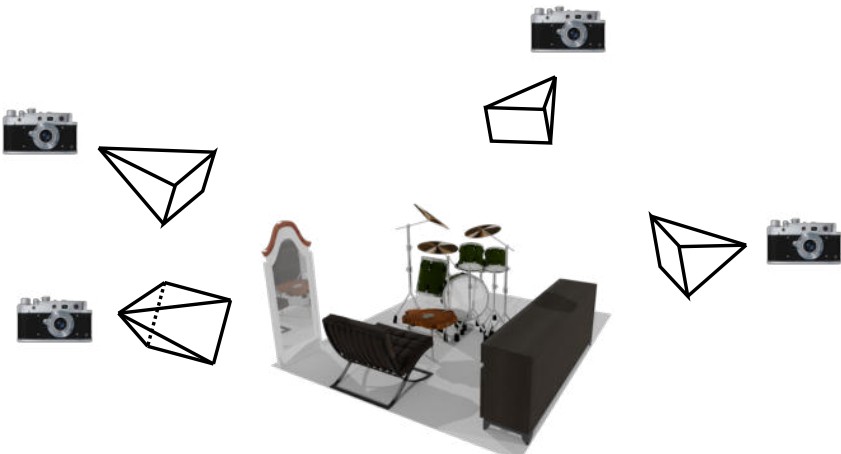

Figure 8: To prevent occlusion between objects from affecting the evaluation of CLIP-Sim, we render one image every 90 degrees of rotation along the outer side of the scene, resulting in a total of 4 rendered images. Finally, the maximum CLIP-Sim value is used as the score for this scene.

failure cases such as incomplete generation or termination due to errors). GSR reflects the generative model's ability to stably produce scenes, which is fundamental for evaluating the reliability of the generation process itself.

$$\text{GSR} = \frac{1}{M} \sum_{m=1}^{M} S_m \tag{3}$$

where $S_m = \mathbb{I}(\cdot)$(the m-th scene is successfully generated) (taking 1 if the scene is generated successfully, 0 otherwise), and M is the total number of generation attempts.

**CLIP-Similarity (CLIP-Sim)** CLIP-Similarity (CLIP-Sim) uses the CLIP model (Contrastive Language-Image Pretraining) to measure semantic similarity between a generated scene and a reference scene/text description. It assesses whether the generated scene aligns with expectations visually or semantically, which is vital for tasks requiring alignment with user intentions or reference images.

For the m-th generated scene, we first calculate its CLIP similarity with the text description from each of the four predefined viewpoints (v=1,2,3,4). The final CLIP similarity for the scene is defined as the maximum value across these four viewpoints:

$$\text{CLIP-Sim}_m = \max_{v \in \{1,2,3,4\}} \left( \frac{\mathbf{f}_{\text{view},m,v} \cdot \mathbf{f}_{\text{text}}}{\|\mathbf{f}_{\text{view},m,v}\| \cdot \|\mathbf{f}_{\text{text}}\|} \right) \tag{4}$$

where $f_{view,m,v}$ is the feature vector of the m-th scene from the v-th viewpoint, $f_{text}$ is the feature vector of the text description, $\cdot$ denotes the vector dot product, and $\|\cdot\|$ denotes the L2 norm of a vector.

### A.4.4 TEXT READABILITY OF USD

The text representation of Universal Scene Description (USD) organizes objects and their transformations (e.g., translation and scaling attributes) in indoor scenes via an ASCII-based structure, ensuring high readability. This feature enables direct editing of object positions and dimensions, facilitating debugging and collaborative design workflows. The corresponding 3D visualization aligns precisely with the text data, demonstrating how USD's structured format underpins the accuracy of spatial mapping and rapid prototyping in 3D scene development—particularly in complex indoor environments.

Beyond boosting the efficiency of manual editing and debugging, this readability also forms a foundation for LLM-driven 3D scene generation. By parsing USD's hierarchical text, LLMs can interpret object relationships and transformation parameters to generate or optimize scene layouts. For ex-

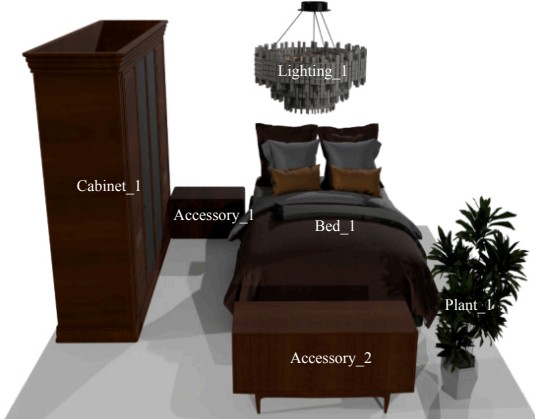

```
1   #usda 1.0
2   ( defaultPrim = "Root" )
3   def "Root" {
4       def "Cabinet_1" (prepend references = @Path/to/Cabinet_1.usdz@) {
5           custom float3 xformOp:rotateXYZ = (0, 0, 0);
6           custom float3 xformOp:scale = (1, 1, 1);
7           custom float3 xformOp:translate = (219.82, 0, 37.46);
8           uniform token[] xformOpOrder = ["xformOp:translate", "xformOp:rotateXYZ", "xformOp:scale"];
9       }
10      def "Bed_1" (prepend references = @Path/to/Bed_1.usdz@) {
11          custom float3 xformOp:rotateXYZ = (180, -90, 180);
12          custom float3 xformOp:scale = (1, 1, 1);
13          custom float3 xformOp:translate = (221.65, 0, 231.01);
14          uniform token[] xformOpOrder = ["xformOp:translate", "xformOp:rotateXYZ", "xformOp:scale"];
15      }
16      def "Accessory_1" (prepend references = @Path/to/Accessory_1.usdz@) {
17          custom float3 xformOp:rotateXYZ = (180, -90, 180);
18          custom float3 xformOp:scale = (1, 1, 1);
19          custom float3 xformOp:translate = (327.74, 0, 109.79);
20          uniform token[] xformOpOrder = ["xformOp:translate", "xformOp:rotateXYZ", "xformOp:scale"];
21      }
22      def "Accessory_2" (prepend references = @Path/to/Accessory_2.usdz@) {
23          custom float3 xformOp:rotateXYZ = (180, 90, 180);
24          custom float3 xformOp:scale = (1, 1, 1);
25          custom float3 xformOp:translate = (23.28, 0, 213.95);
26          uniform token[] xformOpOrder = ["xformOp:translate", "xformOp:rotateXYZ", "xformOp:scale"];
27      }
28      def "Plant_1" (prepend references = @Path/to/Plant_1.usdz@) {
29          custom float3 xformOp:rotateXYZ = (180, 0, 180);
30          custom float3 xformOp:scale = (1, 1, 1);
31          custom float3 xformOp:translate = (139.84, 0, 318.66);
32          uniform token[] xformOpOrder = ["xformOp:translate", "xformOp:rotateXYZ", "xformOp:scale"];
33      }
34      def "Lighting_1" (prepend references = @Path/to/Lighting_1.usdz@) {
35          custom float3 xformOp:rotateXYZ = (0, 0, 0);
36          custom float3 xformOp:scale = (1, 1, 1);
37          custom float3 xformOp:translate = (153.71, 180.152, 210.22);
38          uniform token[] xformOpOrder = ["xformOp:translate", "xformOp:rotateXYZ", "xformOp:scale"];
39      }
40  }
```

Figure 9: Scenes in USD format feature text readability, which facilitates LLM-driven methods to directly generate and edit 3D scenes.

ample, guided by natural language instructions, LLMs can dynamically adjust object positions or introduce new elements, leveraging USD's structure to enable automated scene design.

As shown in Fig. 10, IL3D is adaptable to a range of LLM-driven visual tasks in complex indoor scenes, while also being compatible with multiple simulation platforms. It supports embodiment-related simulation experiments, accelerates model development, enables data-driven generation pipelines, and preserves text readability for further refinement.

### A.5 MULTIMODAL DATA EXPORT

As shown in the Fig. 11, the IL3D dataset supports flexible multimodal data export capabilities, enabling users to extract data tailored to specific task requirements.

In terms of 3D data, it covers core types such as semantic point clouds and 3D bounding boxes of objects. These 3D data retain fine-grained scene geometric and semantic information, providing direct input for

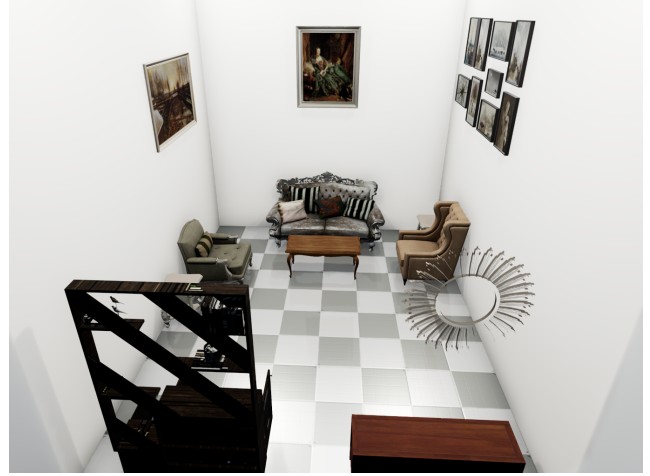

Figure 10: Importing the scenes in IL3D into simulation software can support experiments related to embodied intelligence.

tasks like 3D scene segmentation, object pose estimation, and spatial relationship analysis. For 2D multimodal data, users can export multi-view color images, depth maps, normal maps, and semantic masks by flexibly configuring camera parameters (e.g., viewing angle, resolution, and spatial position). This multi-view setting ensures comprehensive coverage of scene details, avoiding information loss caused by single-view occlusion.

The exported multimodal data adheres to standard data formats, enabling seamless integration with various visual task pipelines—such as using color images and semantic masks for 2D instance segmentation, or combining depth maps with 3D bounding boxes for cross-modal scene understanding. This adaptability makes IL3D a versatile resource for both academic research and industrial application development.

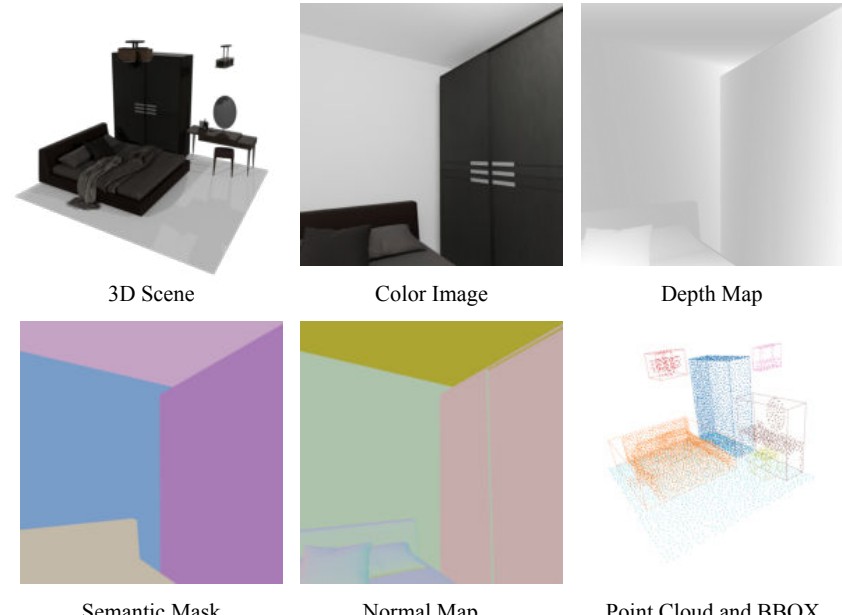

3D Scene         Color Image         Depth Map

Semantic Mask         Normal Map         Point Cloud and BBOX

Figure 11: IL3D can export multiple types of data.

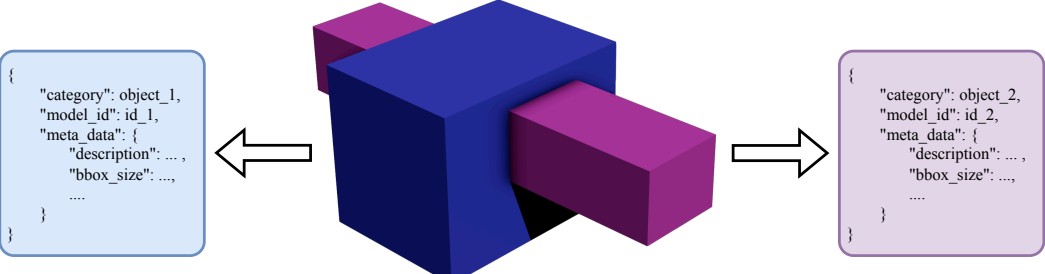

Figure 12: LLM read the information about objects' bounding boxes and their structural descriptions, and their understanding of collisions is also at the bounding box level.

## A.6 ANALYSIS AND DISCUSSION

### A.6.1 TIMING OF OBJECT RETRIEVAL

Existing methods mainly adopt two strategies for object retrieval in scene generation: post-retrieval and pre-retrieval. According to our experiments, the timing of retrieval significantly affects the LLM's reasoning performance for indoor scenes. Post-retrieval involves the model directly predicting a general object distribution; however, without specific object dimensions, this strategy is highly prone to inter-object penetration and objects exceeding room bounding boxes (BBOX).

Pre-retrieval allows the LLM to access detailed information (e.g., dimensions and descriptions) of objects in the scene during reasoning, which improves the quality of generated scenes. For SFT-based methods, however, this requires the model to have a certain level of complex reasoning capability. For models with smaller parameter sizes, introducing pre-retrieval may reduce the performance of some metrics.

### A.6.2 CAN LLMS PERCEIVE FINE-GRAINED OBJECT SHAPES?

As shown in Fig. 12, objects processed by LLM-Driven scene generation methods are represented as annotated BBOXes. Since LLMs cannot perceive the fine-grained surface shapes of objects, they

can only be instructed via prompts to minimize BBOX overlap—but this often leads to physically implausible scenes, such as irrational surface contacts between objects and floating objects.

Existing methods (e.g., InstructScene, MetaScenes) address this issue by sampling point clouds from object surfaces, using pre-trained point cloud encoders to extract 3D shape representations, and inputting these representations into LLMs for reasoning after projection via linear layers; these approaches enable LLMs to perceive fine-grained shapes. Other methods artificially create penetration samples and use Direct Preference Optimization (DPO) to train LLMs to avoid inter-object penetration. While these methods partially resolve the aforementioned problems, abnormal surface contacts between objects remain difficult to completely eliminate.

We argue that a fundamental solution requires designing a method that quantifies inter-object penetration reasonably while ensuring differentiability, and integrating this physical prior into the training of end-to-end generative models. For penetration quantification, mesh-based methods incur excessive computational costs; sampling point clouds with normal directions, however, can balance computational efficiency while preserving topological structures, making it a potentially feasible solution.

### A.6.3 AGENT-BASED VS. SFT-BASED METHODS

Agent-based methods leverage multi-agent systems to repeatedly invoke models for observing and adjusting scene data. Unlike the aforementioned point cloud encoder-based methods, agent-based approaches can render scene images and use vision-language models (VLMs) to observe spatial relationships between objects, enabling high-quality scene generation. Nevertheless, this method has two potential issues:

- **Unstable generation**: Agents may continuously adjust object poses without meeting the termination condition for adjustments, leading to prolonged generation tasks that fail to produce valid scenes and waste significant computational resources.
- **Limited VLM observation**: VLMs struggle to detect objects with abnormal scaling (e.g., an excessively large object enclosing the entire scene and camera). In such cases, VLMs cannot identify the anomaly from rendered images or adjust the object's pose accordingly.

SFT-based methods fine-tune LLMs to learn spatial reasoning capabilities from training data. During reasoning, the model directly accesses object information and performs a limited number of reasoning steps. Although the generated quality is slightly inferior, this approach is more robust and efficient overall.

### A.6.4 LIMITATIONS

While the IL3D dataset provides rich instance-level natural language annotations for 3D scene generation and multimodal learning, it has a limitation: the lack of detailed scene-level descriptions of relationships between objects. In contrast, methods like InstructScene capture spatial and functional associations between objects (e.g., "a lamp on the table" or "a chair near the window") via explicit semantic relationship graphs. IL3D's annotations, however, focus primarily on the attributes and descriptions of individual objects, failing to fully express semantic and topological relationships between objects.

This limitation may restrict the model's performance in complex scene understanding and reasoning tasks—especially in applications like scene editing that require deep contextual associations. We plan to introduce scene-level spatial-semantic relationship annotations in future work to further enhance IL3D's applicability in complex 3D scene generation and interaction tasks.

### A.7 VISUALIZATION AND PROMPTS

We present the 3D scene visualization generated by the LLM in the experiment, as well as the designed prompts.

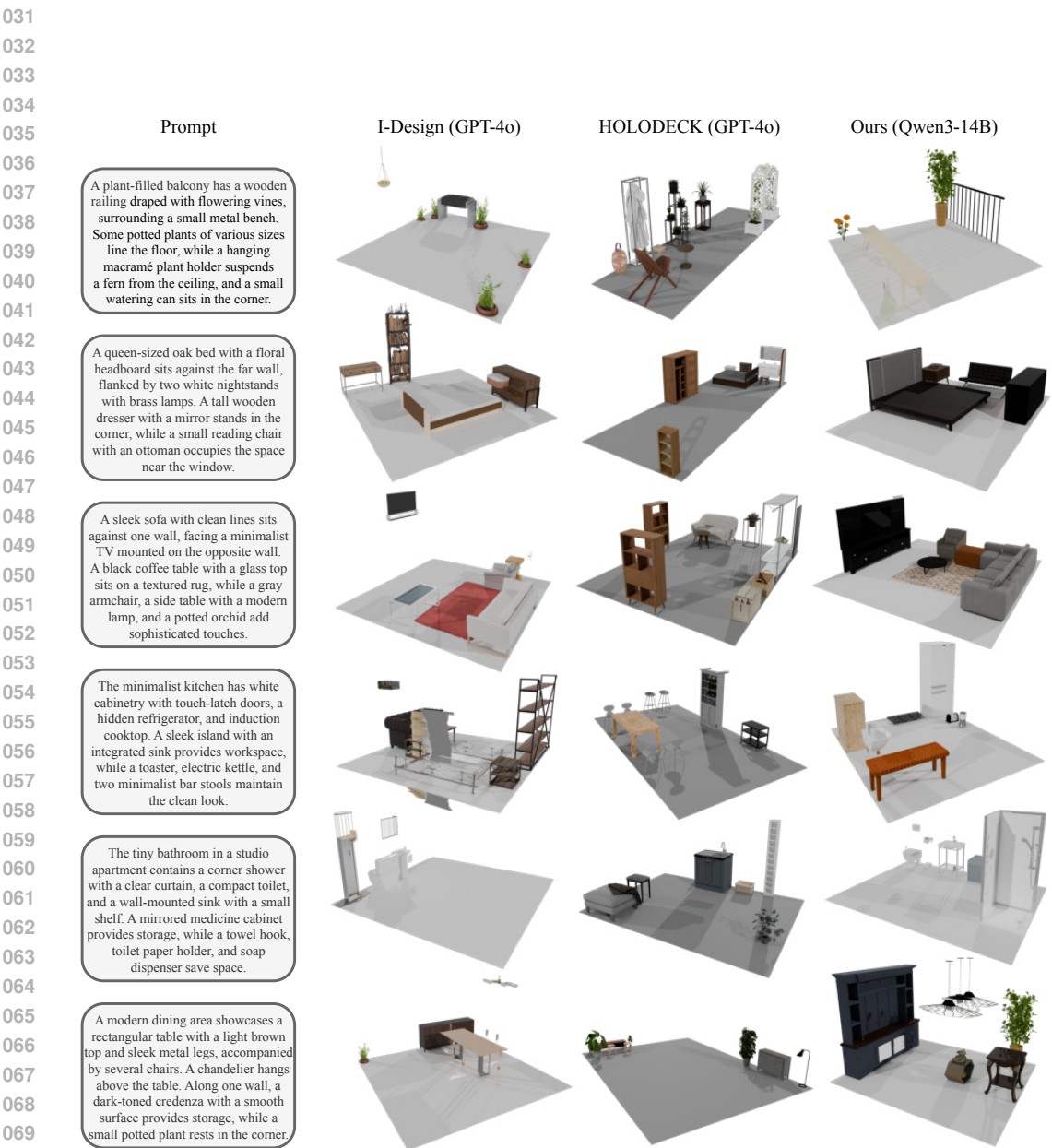

Figure 13: Generated layout comparison across models on I-Design, Holodeck and ours.

**Role**
Your task is to arrange some objects within a given {room_type} effectively.
Follow these guidance to complete your design:

**Rules**
(1) Extract the [Objects] and [Bounding Box Size] from the object information.
(2) Analyze the spatial relationships among [Objects] within the specified [Room Type]. Pay special attention to **avoiding overlap** and **consider other spatial factors like accessibility and aesthetics**.
(3) Determine and design the precise location of all [Objects] ensuring that their bounding boxes do not overlap and that the layout is functional and visually appealing.
(4) I prefer objects to be placed at the edge (the most important constraint) of the room if possible which makes the room look more spacious.
(5) Objects usually need to be aligned in some way (such as parallel or perpendicular to the walls) and **must not extend beyond the floor area**.
(6) Chairs must be placed near to the table/desk and face to the table/desk.
(7) Before specifying the detailed positions of each object, first think about their general arrangement and relative spatial relationships:
a) Which objects need the most space or have fixed positions (like beds, wardrobes)
b) Which objects need to be grouped together (like nightstands with bed)
c) Traffic flow and accessibility considerations.

**Object Information**
{object_information}
*Note: bbox format is [length, width, height] in meters*

**Response Format**
First design the vertices of the floor, then report the 3D spatial coordinates and rotation angles of each object in JSON format, as follows:
{
'Floor': {'xyz': [[8.0, 0, 6.76], [8.0, 0, 0.0], [0.0, 0, 6.76], [0.0, 0, 0.0]]},
'Coffee Tables': [{'position': [1.62, 0.0, 2.29], 'rotation': [180, 90, 180]}],
'Benches': [{'position': [1.72, 0.0, 3.66], 'rotation': [0, 0, 0]}, {'position': [1.63, 0.0, 0.9], 'rotation': [0, 0, 0]}]
}

Important Notes about Coordinate System:
- Y-axis points upward (y=0 is floor level)
- X-axis runs along the room's length from west to east
- Z-axis runs along the room's width from south to north
- All coordinates are in meters
- Output nothing but the JSON (No preamble, no explanation, no additional text of any kind)

Figure 14: The prompts used for scene generation during the training and inference of Qwen3.

**Rules**
(1) Extract the room type from the description (e.g., BedRoom, LivingRoom, Kitchen, etc.).
(2) Identify all mentioned objects and their basic descriptions exactly as described.
(3) If the description mentions multiple instances of the same object, maintain the exact count.
(4) Don't omit the information about the objects in the description.
(5) When encountering quantity words (e.g., six, two, three, multiple) describing objects, split them into individual objects equal to the quantity.
Use the singular form of the object name, and apply the same description to each. For example, "six chairs with blue upholstery" should be split into six separate objects: ["name": "Chair", "description": "A chair with blue upholstery", ...] (repeated six times).
(6) Ensure quantity words are not included in the object name. Focus on the core object name in singular form (e.g., use "Chair" instead of "six chairs" or "Chairs").

**Room Description**
{room_description}

**Response Format**
Report the result of the room_type and dictionaries for each object in JSON format, as follows:
{
'room_type': 'LivingRoom',
'objects': [{'name': 'Sofa', 'A dark green upholstered ottoman with a cushioned lid and decorative brass nailhead trim along its edges.'},
{'name': 'Armchair', 'description': 'A armchair with a sleek design, featuring a cushioned seat and backrest supported by thin metal legs.'},
{'name': 'Armchair', 'description': 'A armchair featuring a curved backrest and armrests. It has a dark green upholstery and thin metal legs.'},
{'name': 'Coffee Table', 'description': 'A modern coffee table with a round wooden top and three sleek legs that taper towards the bottom.'}]
}

Important Notes about response format:
- Output nothing but the JSON. No preamble, no explanation, no additional text of any kind

Figure 15: The prompt used for extracting in-scene information during scene generation.

