# OpenReview forum: "IL3D: A Large-Scale Indoor Layout Dataset for LLM-Driven 3D Scene Generation"
_ICLR.cc/2026/Conference — Submitted to ICLR 2026_

### Official Review · Reviewer_N3uX · 2025-10-27

**Soundness:** 2
**Presentation:** 2
**Contribution:** 1
**Rating:** 2
**Confidence:** 4

**Summary:**

This work presents a new 3D scene dataset (IL3D) for LLM-driven 3D scene generation. This dataset is constructed by integrating, cleaning, and supplementing existing popular datasets (i.e., 3D-FRONT and HSSD) and then adding their own synthetic data to enhance scene diversity and cover underrepresented categories. A key feature is its detailed and USDZ/USDA-format annotations, which were generated using the Qwen3-VL model.

**Strengths:**

- The choice to represent 3D scene data using USDZ-format assets and USDA-format scenes is appropriate. This structure ensures text readability, which is beneficial for both human understanding and direct parsing by LLMs.

- The experiments analyzing the impact of 3D scene dataset scale and quality on LLM scene generation are intriguing. This ablation study provides quantitative data on how training data volume and annotation quality affect key objective metrics (e.g., OOB, OOR, and CLIP-Sim) across different LLM parameter sizes.

**Weaknesses:**

1. **Incomplete Citations and References**. There is a noticeable omission of citations for several works mentioned in the text, such as 3D-FUTURE, InstructScene, and the Qwen3 model family. The current manuscript includes fewer than 30 references, which suggests a failure to fully acknowledge all relevant prior or utilized work. This lack of comprehensive citation undermines the paper's academic rigor and makes it difficult for readers to trace all discussed technologies.

2. **Insufficient Sample Visualization**. Given that the paper's core contribution is the introduction of a new dataset (IL3D), the manuscript provides insufficient illustrative examples of the dataset's samples. While it includes extensive aggregate statistics (e.g., room type distribution, object counts, area analysis ), it lacks rich, individual-sample visualizations that would allow readers to fully grasp the quality, fine-grained nature of the object arrangements, and the detail of the new natural language annotations.

3. **Limited Technical Contribution**. The primary contribution of this work is the dataset itself. However, the dataset's construction mainly involves integrating and cleaning two existing public datasets (3D-FRONT and HSSD) and supplementing them with annotations generated by external models (e.g., Qwen3-VL for instance-level descriptions). As the most crucial bottlenecks in current 3D scene generation datasets are arguably the difficulty of obtaining high-quality 3D assets and realistic object placement schemes, the scope of the novel technical contribution beyond data aggregation and automatic labeling is limited.

**Questions:**

Please refer to the "Weaknesses" section.

---

### Official Review · Reviewer_n89v · 2025-10-31

**Soundness:** 2
**Presentation:** 2
**Contribution:** 2
**Rating:** 2
**Confidence:** 4

**Summary:**

This is a dataset paper. It aims to solve the problem of scarcity of indoor layout dataset. The paper focuses on the method of generating large-scale, high-quality datasets driven by the LLM. The problem of diversity, completeness of annotation and the capability to support multi-modal learning is very severe. IL3d dataset integrates and synthesizes existing ones (3D-FRONT, HSSD) selectively 18 common room types and synthesize many 3d objects as well. It also supports the instance level natural language annotation and multi-modal data outputs. It may support more research in physical-plausible generation, semantic consistent room layouts.

**Strengths:**

Strength
-	The dataset size and annotations are large-scale and dense.
-	The dataset generation methods seem sound.

**Weaknesses:**

-	The core contribution and motivation is to improve the physical plausibility and the semantic coherency in the room layout generation via llm. However, the evaluation of the datasets are limited. E.g. only 2 comparison baseline methods are utilized.

-	The annotation, especially the instance natural language annotation are generated via the multi-view image generation pipeline via an VLM, such QWen3. If so, do the annotations heavily rely on and may generate bias based on this particular VLM. How to evaluate such biases and generate an unbiased dataset for usage? Although in the evaluation, there are quite many evaluation indexes proposed. But none of them tackle the bias of the model.

-	The mechanism and why SFT will enable the reasoning in the 3D space is interesting but confusing. More explanations and visual example comparisons of with and without SFT.

**Questions:**

See weakness

---

### Official Review · Reviewer_EDZ2 · 2025-11-01

**Soundness:** 1
**Presentation:** 1
**Contribution:** 1
**Rating:** 2
**Confidence:** 5

**Summary:**

IL3D positions itself as a large-scale dataset for LLM-driven 3D indoor scene generation: it claims 27,816 layouts (18 room types) and ~29k high-fidelity assets, with instance-level text annotations and multi-modal exports (point clouds, 3D boxes, multi-view RGB, depth, normals, semantic masks). Data are organized in USD (USDZ assets + USDA scenes), with evaluation using OOB/OOR/GSR/CLIP-Sim plus GPT-based scores (OP/PR/SC/SF/VA). The paper compares against I-Design and Holodeck and argues that SFT on IL3D yields more stable objective metrics, with mixed results on subjective scores.
This work does not have novelty, I recommend rejecting it.

**Strengths:**

1. Complete format & export stack: USD organization with one-click export to point cloud/depth/normal/mask is practical and integrable across 2D/3D tasks.

2. Metric coverage: Includes both geometric (OOB/OOR/GSR/CLIP) and GPT-based perceptual metrics, aligning with current literature.

3. Comparisons & ablations present: Shows tables against I-Design/Holodeck and scale/annotation ablations for SFT.

**Weaknesses:**

1. The dataset construction follows exactly the same fundamental pipeline as OptiScene, which is based on the Holodeck and merely scales it up. There is no substantive difference.

2. What can the added modalities (e.g., depth) actually enable downstream? The paper provides no experiments to demonstrate their utility.

3. The writing and figures are rough and poorly executed.

4. The work only releases a dataset; the downstream methods and prompts are almost identical to prior art, with no innovation.

**Questions:**

See weakness

---

### Meta-Review · Area_Chair_f67g · 2025-12-24

**Summary:**

This paper received unanimously negative reviews, and the authors did not submit a rebuttal.

**Reviewer Concerns:**

No rebuttal was submitted. The main concerns raised include similarity with previous work, missing citations, and limited contributions.

**Reviewer Scores:**

No rebuttal was submitted.

---

### Decision · Program_Chairs · 2026-01-26

Reject